# Association study of depressive symptoms and periodontitis in an obese population: Analysis based on NHANES data from 2009 to 2014

**Shuning Li** *, **Jilun Liu, Rui Zhang, Jianfeng Dong**

Department of Oral and Maxillofacial Surgery, The Second Hospital of Hebei Medical University, Shijiazhuang, Hebei, China

* lishuning94@foxmail.com

## Abstract

### Background

Obesity, depressive symptoms, and periodontitis are major worldwide health concerns. Despite separate studies on both illnesses, no research has directly examined the link between depressive symptoms and periodontitis in obese people. Given the close correlation between obesity and chronic illnesses, as well as the possibility of a bidirectional impact between depressive symptoms and periodontitis, this study aims to investigate the link between depressive symptoms and periodontitis in an obese population.

### Methods

This study analyzed data from the NHANES database (2009–2014), including 4,820 persons aged 30 years or older with a BMI over 30. The severity of periodontitis was assessed by clinical attachment loss and probing depth, while symptoms of depressive symptoms were measured using the PHQ-9 questionnaire. After gender, age, race, education, ratio of family income to poverty, sleep duration, diabetes, and cardiovascular illnesses were controlled for, the independent connection between depressive symptoms and periodontitis was investigated using multivariable logistic regression.

### Results

The modified models indicated a significant negative correlation between depressive symptoms and periodontitis (effect size: -0.13, 95% CI: -0.24 to -0.01, p = 0.0266), indicating that depressive symptoms may operate as a protective factor against periodontitis. Interaction studies did not indicate substantial impact modification by factors like age, gender, or education level.

### Conclusion

This study is the first investigation demonstrating a negative link between depressive symptoms and periodontitis in obese persons, indicating a significant interaction between mental

**Data Availability Statement:** The data for our study originates from the National Health and Nutrition Examination Survey (NHANES) database, which is publicly accessible and allows anyone to

access and download the data. It can be retrieved via www.cdc.gov/nchs/nhanes/.

**Funding:** This study was funded by the Health Commission of Hebei Province (20240102). The funders had no role in study design, data collection and analysis, decision to publish, or preparation of the manuscript.

**Competing interests:** The authors have declared that no competing interests exist.

health and dental health in this demographic. The results highlight the importance of comprehensive psychological and oral health care in obese individuals, providing a new avenue for future research and therapeutic applications.

## Introduction

Obesity is defined as a condition in which the body mass index (BMI) exceeds 30 kg/m$^2$. Its prevalence has significantly increased worldwide. In 2008, 1.5 billion adults were classified as overweight, and by 2016, obesity rates had tripled. Obesity is closely linked to multiple chronic diseases, particularly type 2 diabetes [1], cardiovascular diseases, certain cancers [2], hypertension, hyperlipidemia, liver disease, and chronic respiratory diseases. Although obesity is generally regarded as a health risk, some studies have proposed the "obesity paradox," suggesting that obesity may have protective effects in certain contexts [3]. Therefore, assessing the health impact of obesity requires consideration of various factors, including lifestyle, genetics, and environmental influences, making the prevention and management of obesity crucial for public health [4, 5].

Depression is a complex and widespread mental illness, typically defined as a persistent state of low mood accompanied by a significant decrease in interest or pleasure and a drop in energy levels. Symptoms of depression often last for more than two weeks and can considerably impact an individual's daily life and work abilities [6]. The clinical manifestations of depression are diverse and include persistent sadness, anxiety, fatigue, sleep disturbances, changes in appetite, low self-esteem, and suicidal thoughts [7]. According to existing classification systems, depression can be divided into unipolar depression, characterized solely by depressive episodes, and bipolar depression, which includes both depressive and manic or hypomanic symptoms [8]. Depression is closely associated with various medical conditions, such as chronic illnesses (e.g., heart disease, diabetes, and cancer), medication side effects, and genetic factors [9]. Additionally, psychosocial factors such as life stress and interpersonal issues are also considered significant triggers for depression [10, 11]. Therefore, the diagnosis and treatment of depression require a comprehensive consideration of the individual's clinical presentation, medical history, and related factors to develop more effective intervention strategies.

Periodontitis is a chronic inflammatory disease primarily affecting the gums and alveolar bone, potentially leading to tooth mobility and loss, with main symptoms including gum redness, swelling, bleeding, and halitosis. The global prevalence of periodontitis ranges from approximately 9.3% to 11.2% [12] and is closely associated with various chronic systemic diseases, such as type 2 diabetes and cardiovascular diseases [13]. Additionally, factors such as smoking, age, sex, and overweight are significant risk factors influencing its prevalence [14]. These aspects underscore the importance of effective prevention and management of periodontitis to enhance overall oral and systemic health [15].

Currently, there is insufficient evidence regarding the association between periodontal disease and depression in obese individuals, leading to controversy in this area. Existing studies primarily focus on the relationship between dopamine and obesity, without directly exploring the connections among depression, obesity, and periodontal disease [16, 17]. To address this gap, we hypothesize the following: Null Hypothesis: There is no significant association between depressive symptoms and periodontitis in obese individuals. Alternative Hypothesis: There is a significant association between depressive symptoms and periodontitis in obese

individuals. We therefore conducted a detailed study through the NHANES database (2009–2014) to reveal the association between depression and periodontitis in an obese population.

The limitations of this study include the incomplete understanding of the complex interactions between periodontitis, systemic diseases, and biological factors, particularly in obese individuals, where the progression and impact of periodontitis may differ from the general population. Additionally, the study is based on retrospective data and self-reported questionnaires, which may not accurately capture the long-term changes in depressive symptoms and periodontal health and may be subject to reporting bias and the influence of confounding factors, such as medication use or undiagnosed conditions.

## Materials and methods

### Data sources and study population

The National Health and Nutrition Examination Survey (NHANES) gathered information on factors that people were exposed to and the resulting outcomes. This data was collected throughout three consecutive cycles from 2009 to 2014. You may find more information at https://www.cdc.gov/nchs/nhanes. The National Centre for Health Statistics (NCHS) is in charge of NHANES administration. The data were collected through health interviews conducted at participants' residences, health checks carried out at mobile testing centres (MECs), and laboratory samples. There was no need for a supplementary ethical review for this manuscript, as NHANES underwent an ethical review by the National Ethical Review Board for Research in Health Statistics [18].

The study, which initially comprised 30,468 participants (NHANES 2009 through 2014), incorporated the most recent periodontal examination data for US adults. The eligibility of a subject to participate was determined based on the following criteria: (1) NHANES participants aged thirty years or older; (2) individuals who had undergone an oral periodontal examination; and (3) participants in NHANES with a BMI index greater than 30. (4) participants in NHANES with complete depression assessment results. In the end, a grand total of 4820 individuals were enrolled in the research. Fig 1 depicts the procedure of data filtration.

### Outcome variable

Clinical attachment loss (AL) and periodontal pocket probing depth (PD) are typical indicators for evaluating the severity of periodontitis [19] The periodontist examined the 28 eligible participants' teeth at the mobile examination center, explored each tooth at six locations, but did not include the third molars [20]. This study employed the CDC/AAP classification criteria for periodontitis, which were created by Eke et al. in 2012 [21]. 2 neighboring AL $\geq$ 3 mm, $\geq$ 2 neighboring PD $\geq$ 4 mm (not on the same tooth), or 1 neighboring PD $\geq$ 5 mm were the criteria for mild periodontitis. A periodontitis that was classified as moderate was defined as attachment loss (AL) that was either $\geq$ 4 mm (not on the same tooth) or $\geq$ 2 mm (on the same tooth). Two neighboring sites AL $\geq$ 6 mm (not on the same tooth) and at least one neighboring site PD $\geq$ 5 mm were used to designate severe periodontitis. As per the aforementioned criteria, individuals are classified as having mild periodontitis, moderate periodontitis, severe periodontitis, or no periodontitis.

### Exposure variable

The PHQ-9, which is a questionnaire based on self-reporting by patients, was used to assess depressive symptoms. It contains nine items (depressed mood, appetite problems, fatigue, sleep difficulties, psychomotor retardation or agitation, concentration problems, lack of

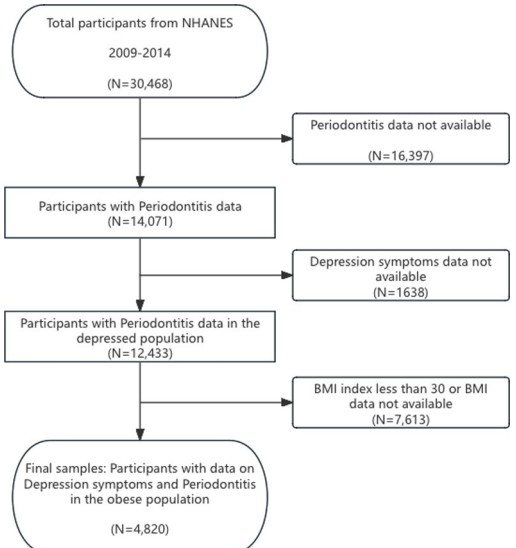

**Fig 1. Flowchart of participant selection.** NHANES, National Health and Nutrition Examination Survey; BMI, Body Mass Index.

interest, feelings of worthlessness, and suicidal ideation) that assess the frequency of depressive symptoms and is well accepted as an accurate and reliable method for depressive symptoms screening [22–25]. The total PHQ-9 score varies from 0 to 27; a score of 10 or greater was defined as indicative of depressive symptoms.

## Confounding variables

Covariates were chosen in accordance with clinical expertise and prior research. This study included the following covariates: gender; age; race; education level; ratio of family income to poverty(RIP); sleep time(hours); sleep disorder; diabetes; stroke; hypertension; coronary heart disease; cigarette use; alcohol use. Gender was classified as either male or female, while age was segmented into four categories based on quartiles. The race categories included Mexican American, Other Hispanic, Non-Hispanic White, Non-Hispanic Black, and Other Race. Educational level was categorized into five categories: Less Than 9th Grade, 9-11th Grade (including 12th grade with no diploma), High School Grad/GED or Equivalent, Some College or AA degree, and College Graduate or above. Diabetes status was categorized into three groups: YES, NO, or Borderline, based on whether a doctor had diagnosed the participant with diabetes. Cigarette use was categorized as never smoker (smoked < 100 cigarettes in lifetime), former smoker (smoked > 100 cigarettes but currently quit smoking), current smoker (smoked > 100 cigarettes and currently smoking) [26]. According to past research [27], we defined moderate drinking as 14 or fewer drinks/week for men or 7 or fewer drinks/week for women or 5 or fewer drinks/day on any single day in the past year for both men and women. Similarly, we defined heavy drinking as more than 14 drinks/week for men or more than 7 drinks/week for women or 5 or more drinks/day on any single day at least once in the past year for either men or women.

## Statistical analysis

The statistical analysis of the results was performed using the R software (R4.2.1, http://www.R-project.org) in combination with Empower Stats 4.1, which could be accessed at http://

www.empowerstats.com. The Kruskal-Wallis H-test was used to evaluate statistical disparities among different groups (quartiles) for continuous variables, while the chi-square test was performed for categorical variables. The study utilized multivariate logistic regression to investigate the independent association between depressive symptoms and periodontitis. Three models were employed: an unadjusted model, a model adjusted for gender; age; race; education level; RIP (model 1), and a model adjusted for gender; age; race; education level; RIP; sleep time(hours); sleep disorder; diabetes; stroke; hypertension; coronary heart disease; cigarette use; alcohol use (model 2). The robustness of the results was guaranteed by employing trend p values as sensitivity analyses [28]. Analyzed subgroups and conducted interaction tests to assess the influence of variables on the relationship between depressive symptoms and periodontitis [29].

## Results

In this study, we analyzed a total of 12,433 participants with complete data on periodontitis and depressive symptoms. The results revealed a significant difference in the prevalence of periodontitis between the depressive and non-depressive groups. Specifically, the prevalence of periodontitis was 42.01% in the depression group, significantly lower than the 57.99% observed in the non-depression group (P = 0.039). Additionally, significant differences were noted in other characteristics; the depression group had a higher proportion of females (65.44%, P < 0.001) and a greater percentage of individuals with education levels of junior high school or below (15.64%, P < 0.001). The rates of hypertension, diabetes, and stroke were also significantly higher in the depression group compared to the non-depression group (P < 0.001) (Table 1).

The results of this study highlight a significant negative association between depressive symptoms and periodontal disease. The unadjusted analysis showed an effect estimate of -0.06 (95% CI: -0.15, 0.03), which did not reach statistical significance (p = 0.1612). However, in the Adjusted I model, the effect estimate improved significantly to -0.14 (95% CI: -0.23, -0.04), with a p-value of 0.0037, indicating a significant negative association between depressive symptoms and periodontal disease. In the Adjusted II model, the effect estimate remained at -0.13 (95% CI: -0.24, -0.01), with a p-value of 0.0266, further confirming this relationship. These findings underscore the importance of considering depressive symptoms as a potential factor in the pathophysiology of periodontal disease, suggesting that clinical practice should address the impact of mental health on oral health (Table 2).

In the interaction analysis, we examined the relationship between depressive symptoms and periodontitis in the context of various effect modifiers, considering age, gender, education level, race, alcohol use, cigarette use, diabetes, and sleep time (hours) as influencing factors. We found that all p-values were greater than 0.05, indicating that no significant changes were observed (Table 3).

## Discussion

The present study primarily explores the relationship between depression and periodontal disease in obese populations. Through a large sample analysis using the NHANES database, we found that, in a fully adjusted model, depression was significantly negatively correlated with periodontal disease, with an effect size of -0.13 (-0.24, -0.01), p = 0.0266.

This study investigates the relationship between depression and periodontal disease in obese individuals, and currently, there is no related research available. This study fills a gap in this area.

**Table 1. Characterization of the study population based on depressive symptoms.**

| Depressive symptoms | NO | YES | Standardize diff. | P-value | P-value* |
|---|---|---|---|---|---|
| N | 11212 | 1221 | | | |
| AGE | 54.62 ± 14.96 | 53.48 ± 13.29 | 0.08 (0.02, 0.14) | 0.011 | 0.023 |
| Ratio Of Family Income To Poverty | 2.68 ± 1.64 | 1.65 ± 1.33 | 0.69 (0.63, 0.76) | <0.001 | <0.001 |
| Sleep Time (hours) | 6.88 ± 1.37 | 6.33 ± 1.86 | 0.33 (0.27, 0.39) | <0.001 | <0.001 |
| Gender | | | 0.33 (0.27, 0.39) | <0.001 | - |
| male | 5697 (50.81%) | 422 (34.56%) | | | |
| female | 5515 (49.19%) | 799 (65.44%) | | | |
| Race | | | 0.18 (0.12, 0.24) | <0.001 | - |
| Mexican American | 1496 (13.34%) | 180 (14.74%) | | | |
| Other Hispanic | 1047 (9.34%) | 166 (13.60%) | | | |
| Non-Hispanic White | 5110 (45.58%) | 531 (43.49%) | | | |
| Non-Hispanic Black | 2338 (20.85%) | 256 (20.97%) | | | |
| Other Race | 1221 (10.89%) | 88 (7.21%) | | | |
| Education Level | | | 0.45 (0.39, 0.51) | <0.001 | - |
| Less Than 9th Grade | 1109 (9.90%) | 191 (15.64%) | | | |
| 9-11th Grade | 1549 (13.83%) | 276 (22.60%) | | | |
| High School Grad/GED or Equivalent | 2487 (22.21%) | 275 (22.52%) | | | |
| Some College or AA degree | 3130 (27.95%) | 345 (28.26%) | | | |
| College Graduate or above | 2924 (26.11%) | 134 (10.97%) | | | |
| Stroke | | | 0.19 (0.13, 0.25) | <0.001 | - |
| yes | 422 (3.77%) | 102 (8.37%) | | | |
| no | 10780 (96.23%) | 1117 (91.63%) | | | |
| Hypertension | | | 0.28 (0.23, 0.34) | <0.001 | - |
| yes | 4641 (41.44%) | 677 (55.54%) | | | |
| no | 6557 (58.56%) | 542 (44.46%) | | | |
| Diabetes | | | 0.26 (0.20, 0.32) | <0.001 | - |
| yes | 1569 (14.00%) | 293 (24.06%) | | | |
| no | 9318 (83.14%) | 888 (72.91%) | | | |
| borderline | 321 (2.86%) | 37 (3.04%) | | | |
| Coronary Heart Disease | | | 0.13 (0.07, 0.19) | <0.001 | - |
| yes | 493 (4.41%) | 90 (7.46%) | | | |
| no | 10688 (95.59%) | 1117 (92.54%) | | | |
| Cigarette Use | | | 0.40 (0.34, 0.46) | <0.001 | - |
| never smoker | 6101 (54.44%) | 493 (40.38%) | | | |
| former smoker | 3073 (27.42%) | 301 (24.65%) | | | |
| current smoker | 2033 (18.14%) | 427 (34.97%) | | | |
| Alcohol Use | | | 0.15 (0.08, 0.22) | <0.001 | - |
| never drinking | 1598 (17.74%) | 169 (18.95%) | | | |
| moderate drinking | 7022 (77.94%) | 655 (73.43%) | | | |
| heavy drinking | 389 (4.32%) | 68 (7.62%) | | | |
| Sleep Disorder | | | 0.52 (0.46, 0.58) | <0.001 | - |
| yes | 910 (8.13%) | 332 (27.30%) | | | |
| no | 10289 (91.87%) | 884 (72.70%) | | | |
| Periodontitis | | | 0.06 (0.00, 0.12) | 0.039 | - |
| no | 6155 (54.90%) | 708 (57.99%) | | | |
| yes | 5057 (45.10%) | 513 (42.01%) | | | |

Results in table: Mean + SD / N (%).

P-value*: obtained by Kruskal Wallis rank sum test for continuous variables, and Fisher's exact probability test for count variables with theoretical number < 10.

**Table 2. The relationship between depressive symptoms and periodontitis in individuals with obesity.**

| Exposure | Non-adjusted | Adjusted Model I | Adjusted Model II |
|---|---|---|---|
| Depressive Symptoms | | | |
| no | 0 | 0 | 0 |
| yes | -0.06 (-0.15, 0.03) 0.1612 | -0.14 (-0.23, -0.04) 0.0037 | -0.13 (-0.24, -0.01) 0.0266 |

Results in table:β (95% CI) P-value / OR (95% CI) P-value

Outcome Variable: Periodontitis

Exposure Variable: Depressive Symptoms

Non-adjusted Model: Adjusted for none.

Adjusted Model I: Adjusted for gender, race, ratio of family income to poverty, education level, and age (quartiles).

Adjusted Model II: Adjusted for gender, age (quartiles), race, education level, ratio of family income to poverty, stroke, sleep time (hours), sleep disorder, diabetes, coronary heart disease, cigarette use, alcohol use, and hypertension.

The relationship between periodontitis and obesity is significant, with obesity recognized as an important risk factor for periodontitis. This relationship is mediated through various mechanisms, including inflammatory responses (the secretion of pro-inflammatory cytokines and insulin resistance induced by obesity) [30], oxidative stress (the increase of reactive oxygen species negatively impacting periodontal health) [31], changes in microbial communities (the increased levels and proportions of pathogens associated with periodontal disease in obese individuals) [32], and metabolic disorders (such as diabetes and cardiovascular diseases affecting the immune system) [33]. These mechanisms interact to promote the association between obesity and periodontitis. Therefore, it is crucial to consider the obesity status of patients and adopt corresponding preventive and therapeutic measures in the treatment of periodontal disease [34].

The relationship between periodontitis and depression is complex and multifaceted, with numerous studies demonstrating a positive correlation between the two. Epidemiological data indicate that individuals with periodontitis have a significantly increased risk of subsequent development of depression, suggesting that the existence of periodontitis may contribute to psychological distress, such as anxiety and depression, often stemming from pain and the stress of treatment [35, 36]. Additionally, dysregulation of the immune system and changes in the microbiome are believed to play crucial roles in this association, with specific oral pathogens potentially influencing emotional and cognitive functions through inflammatory responses [37, 38]. From a neurobiological perspective, brain-derived neurotrophic factor (BDNF) has been shown to exhibit depressive-like behavior in animal models of periodontitis, indicating that neurobiological pathways may be involved in linking these conditions [39]. While most studies support the notion that periodontitis affects depression, there is also evidence suggesting that depression may increase the risk of periodontitis, indicating a possible bidirectional relationship [40]. Therefore, further research is needed to elucidate the causal relationships and biological mechanisms to gain a deeper understanding of the interactions between these two diseases.

The interplay between periodontitis, obesity, and depression is multifaceted, with a positive correlation observed between periodontitis and obesity, likely due to inflammatory mediators and altered adipokine profiles that exacerbate periodontal tissue destruction [15, 33]. However, within obese populations, a negative correlation between periodontitis and depression has been noted, potentially attributable to several mechanisms. Obesity can lead to significant alterations in the levels of adipokines, such as adiponectin and leptin. These changes in adipokine levels may not only alleviate the severity of periodontitis through anti-inflammatory

**Table 3. The analysis of the relationship between depressive symptoms and periodontitis in obese populations under the influence of different effect modifiers.**

| Exposure: Depressive Symptoms | Effect modifier | | N | Statistics | Model II* | P for interaction |
|---|---|---|---|---|---|---|
| no | Age (quartiles) | ageQ1 | 996 | 996 (72.89) | Ref. | 0.5443 |
| yes | | ageQ1 | 130 | 130 (66.92) | -0.07 (-0.28, 0.14) 0.5164 | |
| no | | ageQ2 | 1077 | 1077 (103.34) | 0.12 (-1.48, 1.73) 0.8819 | |
| yes | | ageQ2 | 181 | 181 (94.48) | -0.08 (-1.69, 1.52) 0.9176 | |
| no | | ageQ3 | 995 | 995 (114.87) | -0.24 (-1.74, 1.27) 0.7589 | |
| yes | | ageQ3 | 201 | 201 (98.01) | -0.42 (-1.94, 1.09) 0.5825 | |
| no | | ageQ4 | 1130 | 1130 (87.35) | -0.43 (-1.92, 1.06) 0.5717 | |
| yes | | ageQ4 | 110 | 110 (84.55) | -0.42 (-1.91, 1.07) 0.5802 | |
| no | Education Level | Less Than 9th Grade | 417 | 417 (118.94) | Ref. | 0.5235 |
| yes | | Less Than 9th Grade | 98 | 98 (104.08) | -0.21 (-0.54, 0.11) 0.1998 | |
| no | | 9-11th Grade | 613 | 613 (112.23) | 0.21 (-1.02, 1.45) 0.7338 | |
| yes | | 9-11th Grade | 139 | 139 (84.89) | 0.11 (-1.13, 1.35) 0.8596 | |
| no | | High School Grad/GED or Equivalent | 998 | 998 (104.01) | 0.23 (-0.90, 1.37) 0.6873 | |
| yes | | High School Grad/GED or Equivalent | 142 | 142 (87.32) | -0.03 (-1.17, 1.11) 0.9579 | |
| no | | Some College or AA degree | 1322 | 1322 (86.84) | -0.17 (-1.27, 0.94) 0.7683 | |
| yes | | Some College or AA degree | 176 | 176 (88.64) | -0.20 (-1.30, 0.89) 0.7161 | |
| no | | College Graduate or above | 845 | 845 (70.41) | -0.44 (-1.67, 0.79) 0.4798 | |
| yes | | College Graduate or above | 67 | 67 (71.64) | -0.43 (-1.69, 0.82) 0.4988 | |
| no | Gender | male | 1950 | 1950 (111.13) | Ref. | 0.1625 |
| yes | | female | 178 | 178 (112.92) | -0.00 (-0.20, 0.20) 0.9707 | |
| no | | male | 2248 | 2248 (80.16) | 0.48 (-0.17, 1.12) 0.1455 | |
| yes | | female | 444 | 444 (78.15) | 0.30 (-0.34, 0.95) 0.3511 | |
| no | Race | Mexican American | 646 | 646 (122.45) | Ref. | 0.6198 |
| yes | | Mexican American | 109 | 109 (107.34) | -0.07 (-0.34, 0.20) 0.6024 | |
| no | | Other Hispanic | 390 | 390 (101.03) | -0.29 (-1.68, 1.09) 0.6792 | |
| yes | | Other Hispanic | 82 | 82 (82.93) | -0.61 (-1.99, 0.76) 0.3818 | |
| no | | Non-Hispanic White | 1840 | 1840 (77.72) | -0.37 (-1.43, 0.69) 0.4962 | |
| yes | | Non-Hispanic White | 263 | 263 (72.24) | -0.46 (-1.52, 0.61) 0.4010 | |
| no | | Non-Hispanic Black | 1131 | 1131 (104.60) | -0.41 (-1.56, 0.75) 0.4900 | |
| yes | | Non-Hispanic Black | 138 | 138 (105.07) | -0.40 (-1.56, 0.76) 0.4988 | |
| no | | Other Race | 191 | 191 (89.53) | -0.34 (-2.30, 1.62) 0.7351 | |
| yes | | Other Race | 30 | 30 (93.33) | -0.46 (-2.34, 1.43) 0.6353 | |
| no | Alcohol Use | never drinking | 625 | 625 (91.68) | Ref. | 0.8795 |
| yes | | never drinking | 91 | 91 (95.60) | -0.12 (-0.38, 0.13) 0.3359 | |
| no | | moderate drinking | 2528 | 2528 (94.42) | 0.15 (-0.61, 0.92) 0.6962 | |
| yes | | moderate drinking | 322 | 322 (88.20) | 0.03 (-0.73, 0.80) 0.9306 | |
| no | | heavy drinking | 138 | 138 (85.51) | -0.94 (-3.01, 1.14) 0.3769 | |
| yes | | heavy drinking | 31 | 31 (87.10) | -0.93 (-2.96, 1.10) 0.3708 | |
| no | Cigarette Use | never smoker | 2369 | 2369 (86.53) | Ref. | 0.6472 |
| yes | | never smoker | 276 | 276 (82.25) | -0.09 (-0.24, 0.07) 0.2898 | |
| no | | former smoker | 1195 | 1195 (95.48) | -0.15 (-0.87, 0.58) 0.6888 | |
| yes | | former smoker | 161 | 161 (85.71) | -0.25 (-0.99, 0.48) 0.4973 | |
| no | | current smoker | 632 | 632 (122.78) | 0.52 (-0.52, 1.55) 0.3274 | |
| yes | | current smoker | 185 | 185 (98.92) | 0.30 (-0.73, 1.34) 0.5657 | |
| no | Diabetes | yes | 855 | 855 (101.40) | Ref. | 0.9747 |
| yes | | yes | 202 | 202 (89.11) | -0.11 (-0.33, 0.10) 0.3089 | |
| no | | no | 3176 | 3176 (92.16) | 0.30 (-0.42, 1.01) 0.4146 | |

*(Continued)*

**Table 3.** (Continued)

| Exposure: Depressive Symptoms | | Effect modifier | N | Statistics | Model II* | P for interaction |
|---|---|---|---|---|---|---|
| yes | | no | 396 | 396 (89.14) | 0.18 (-0.53, 0.89) 0.6146 | |
| no | | borderline | 166 | 166 (104.22) | -1.07 (-2.68, 0.54) 0.1916 | |
| yes | | borderline | 23 | 23 (65.22) | -1.25 (-2.88, 0.38) 0.1326 | |

Results in table: β (95% CI) P value / OR (95% CI) P value

Outcome variable: Periodontitis

Exposure variable: Depressive Symptoms

pathways [41] but also regulate neuroinflammatory responses in the brain [42], thereby reducing depressive symptoms [43–45]; dysfunction of the hypothalamic-pituitary-adrenal (HPA) axis induced by chronic metabolic stress is significantly associated with depression [46]. However, in certain circumstances, this adaptive mechanism may mitigate the psychological stress caused by metabolic disturbances, thereby reducing depressive symptoms [47]; and lifestyle factors, such as adopting a healthier diet, increasing physical activity, and enhancing social engagement, that contribute to improved mental health [48]. While these mechanisms provide a plausible explanation for the observed negative correlation, it is important to note that the above hypotheses are primarily based on existing literature and theoretical reasoning. Given that this study relies on retrospective data and patient-reported questionnaires, it is not possible to directly validate these assumptions. Therefore, further longitudinal and experimental studies are necessary to clarify the complex interactions between obesity, periodontitis, and depression.

This study found a negative correlation between depression and periodontitis in obese individuals, suggesting a complex interplay between mental and physical health. This finding underscores the importance of integrated management of psychological and oral health in the assessment and treatment of obese patients. Early intervention and patient education may improve overall health. Furthermore, this discovery provides a new direction for future research to explore the underlying mechanisms linking obesity, depression, and periodontitis.

This study has several strengths, including the use of NHANES data with a large sample size that enhances the reliability of the results. It also uniquely focuses on the relationship between periodontal disease and depression in obese populations, providing a new perspective in the field, with clinically relevant evidence. However, the study also has notable limitations. The findings are only applicable to obese individuals and are restricted to the U.S. population, making it difficult to generalize to other populations. Additionally, as an observational study, it can only establish an association between the two conditions without proving causation, and it did not control for all potential confounding factors. Moreover, the use of the PHQ-9 scale, which measures symptoms over the past two weeks, may lead to misclassification by capturing transient emotional states rather than chronic depression. While efforts were made to account for various confounding factors, the possibility remains that unaccounted confounders could affect the relationship between depressive symptoms and periodontitis in an obese population. Therefore, caution is warranted in interpreting the results, and further research is needed for validation.

## Conclusion

This study, based on NHANES data from 2009 to 2014, is the first to reveal a significant negative correlation between depressive symptoms and periodontal disease in obese populations,

suggesting that depressive symptoms may act as a protective factor. The findings highlight the importance of integrated management of mental and oral health in obese patients. Despite certain limitations, particularly regarding the generalizability of the results, future research should further explore the mechanisms linking obesity, depressive symptoms, and periodontal disease, and validate these findings in diverse populations.

## Author Contributions

**Conceptualization:** Shuning Li.

**Data curation:** Shuning Li, Rui Zhang.

**Formal analysis:** Shuning Li, Rui Zhang.

**Funding acquisition:** Jilun Liu.

**Investigation:** Shuning Li.

**Methodology:** Shuning Li, Jianfeng Dong.

**Project administration:** Shuning Li.

**Resources:** Shuning Li.

**Software:** Jianfeng Dong.

**Supervision:** Shuning Li.

**Validation:** Shuning Li, Jianfeng Dong.

**Visualization:** Rui Zhang.

**Writing – original draft:** Shuning Li, Rui Zhang.

**Writing – review & editing:** Shuning Li.

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
