## [Decision Letter · Decision Letter 0]

20 Nov 2024

PONE-D-24-44796Association Study of Depression and Periodontitis in an Obese Population: Analysis Based on NHANES Data from 2009 to 2014PLOS ONE

Dear Dr. Li,

Thank you for submitting your manuscript to PLOS ONE. After careful consideration, we feel that it has merit but does not fully meet PLOS ONE’s publication criteria as it currently stands. Therefore, we invite you to submit a revised version of the manuscript that addresses the points raised during the review process.

We look forward to receiving your revised manuscript.

Kind regards,

Isabel Cristina Gonçalves Leite

Academic Editor

PLOS ONE

Journal Requirements:

“This study was funded by the Health Commission of Hebei Province (20240102).”

3. Please note that your Data Availability Statement is currently missing the repository name. If your manuscript is accepted for publication, you will be asked to provide these details on a very short timeline. We therefore suggest that you provide this information now, though we will not hold up the peer review process if you are unable.

4. We note you have included a table to which you do not refer in the text of your manuscript. Please ensure that you refer to Table 1 in your text; if accepted, production will need this reference to link the reader to the Table.

**Additional Editor Comments:**

Need to comply with reviewers’ analyses to request publication.

Reviewers' comments:

Reviewer's Responses to Questions

**Comments to the Author**

1. Is the manuscript technically sound, and do the data support the conclusions?

Reviewer #1: Yes

Reviewer #2: Yes

2. Has the statistical analysis been performed appropriately and rigorously? 

Reviewer #1: Yes

Reviewer #2: I Don't Know

3. Have the authors made all data underlying the findings in their manuscript fully available?

Reviewer #1: Yes

Reviewer #2: Yes

4. Is the manuscript presented in an intelligible fashion and written in standard English?

Reviewer #1: Yes

Reviewer #2: Yes

5. Review Comments to the Author

Reviewer #1: This manuscript examines the association between depressive symptoms and periodontitis in an obese population, using NHANES data (2009-2014). The study reveals an unexpected finding: a negative association between depressive symptoms and periodontitis, suggesting that depressive symptoms might act as a protective factor against periodontitis in obese individuals. This result diverges from most existing research, which links depression with an increased risk of periodontitis, likely due to behavioral and biological mechanisms such as systemic inflammation and reduced motivation for self-care.

Major Points

1. Explanation of Unexpected Results:

o The authors attempt to explain the inverse association observed by proposing mechanisms unique to obesity. These include metabolic and inflammatory pathways influenced by adipokines, as well as behavioral coping factors. While these suggestions are interesting, they are largely speculative and would benefit from further empirical support or a more in-depth review of relevant literature.

o The manuscript would be strengthened by a deeper exploration of how obesity uniquely alters the inflammatory and immune response, particularly focusing on how these changes might interact with depression to impact periodontal health. Specific references to literature on adipokines (e.g., leptin, adiponectin) and systemic inflammatory markers would provide a stronger foundation for the proposed mechanisms.

2. Consideration of Additional Comparison Groups:

o The study’s insights would be enhanced by including additional comparison groups, specifically a depressed, non-obese group and a non-depressed, non-obese group. This addition would allow for a clearer comparison and could help determine if the negative association between depressive symptoms and periodontitis is unique to the obese population or is an artifact of the study design.

o Including these groups would also help contextualize the findings within broader population health and clarify whether obesity alone accounts for the protective association observed.

3. Terminology Consistency:

o The study relies on the PHQ-9 to measure depressive symptoms, which captures recent emotional states rather than diagnosing chronic or clinical depression. Thus, it is crucial to refer consistently to participants as having "depressive symptoms" rather than as being “depressed.” Using “depressed” might imply a clinical diagnosis, which the PHQ-9 does not provide. Maintaining consistent terminology would more accurately reflect the study's methodology and prevent potential misinterpretation of the findings.

4. Limitations of the PHQ-9 Questionnaire:

o The PHQ-9 reflects recent symptoms (over the past two weeks), and thus might capture transient emotional states rather than chronic depressive disorders. This could lead to misclassification, as some individuals scoring high might not have clinical depression but could be experiencing temporary stress or emotional fluctuations. Acknowledging this limitation in the manuscript would enhance transparency and help readers interpret the findings cautiously. For future studies, the authors could consider using longitudinal measures or clinical diagnostic tools to distinguish between temporary and chronic depressive conditions.

5. Context in Existing Literature:

o The manuscript would benefit from an expanded introduction to contextualize the effects of both obesity and depression on periodontitis individually.

Obesity and Periodontitis: The literature generally supports a positive association between obesity and periodontitis due to factors like systemic inflammation, insulin resistance, and immune dysfunction caused by excess adipose tissue. These conditions contribute to periodontal tissue degradation, increasing periodontitis risk. A more detailed background on obesity’s direct influence on periodontitis would strengthen the reader’s understanding of why this population might experience unique associations with depressive symptoms and periodontal health.

Depression and Periodontitis: Similarly, depression is known to be associated with a higher risk of periodontitis. This link is commonly attributed to decreased motivation for oral hygiene, increased inflammation, and immune dysregulation associated with depressive states. These insights would better frame the current study’s novelty and emphasize the surprising nature of the observed negative association in an obese population.

6. Behavioral Factors and Health-Seeking Assumptions:

o The authors suggest that depressed individuals may engage in certain coping mechanisms or seek mental health support, indirectly benefiting their oral health. However, these behavioral assumptions are not directly supported by data within the study, as no information on coping strategies or health behaviors was collected. If the authors are to suggest behavioral coping as a factor, future studies should collect data on participants’ health-seeking behaviors, mental health management, and oral hygiene practices to substantiate this claim.

7. Unexplored Biological Pathways:

o While the authors suggest that metabolic changes in obese individuals might alter inflammatory responses to depressive symptoms, this hypothesis is not directly tested in the study. The inclusion of specific physiological markers, such as cortisol levels, cytokines, or adipokines, could have provided more concrete insights into how depressive symptoms interact with obesity-related inflammatory processes to impact periodontal health. Future studies that incorporate these biomarkers would enable a more direct investigation into the proposed mechanisms.

Strengths:

• Novel Research Focus: This study addresses a previously unexplored association between depression and periodontitis within an obese population, filling a gap in the literature and opening avenues for further research.

• Large Sample Size: The NHANES data provides a large, representative sample, which enhances the reliability and statistical power of the findings.

• Comprehensive Covariate Adjustment: The use of multivariable logistic regression with numerous covariates (age, race, education, etc.) strengthens the credibility of the findings by minimizing confounding effects.

Weaknesses:

• Speculative Explanations: The biological and behavioral explanations provided are plausible but lack empirical support, as no biomarkers or behavioral data were collected to substantiate these claims.

• Inadequate Exploration of Relevant Pathways: More depth in discussing obesity’s effects on immune function and inflammation would provide a stronger theoretical basis for the study’s findings.

• Behavioral Assumptions without Supporting Data: The authors suggest coping behaviors as a factor in the observed association but lack direct data on health-seeking behaviors, mental health support, or oral hygiene practices, weakening this explanation.

• Terminology and Measurement Limitations: The reliance on PHQ-9 without clinical diagnosis could lead to misclassification, as it captures recent depressive symptoms rather than chronic depression. Consistent use of “depressive symptoms” is recommended for clarity.

This study provides valuable insights into the relationship between depressive symptoms and periodontitis in an obese population, challenging established assumptions. While the findings are intriguing, the speculative nature of the explanations and lack of additional comparison groups limit the interpretability. Future research should:

• Incorporate non-obese comparison groups to assess whether the observed associations are specific to obesity.

• Use longitudinal designs to explore the stability and causality of these associations over time.

• Collect biomarker data and behavioral information to substantiate proposed biological and psychological mechanisms.

Addressing these areas in future studies will enhance the rigor and applicability of the findings, contributing to a more nuanced understanding of how mental health and periodontal health intersect within unique populations.

Reviewer #2: Interesting results, definitely not something we are used to see in terms of the results and negative correlation of periodontitis and depression. Please see below, some of my comments.

Introduction:

• “and its prevalence has significantly increased globally, with 1.5 billion adults being overweight in 2008 and obesity rates tripling by 2016.” Long sentence and the information seem redundant. Consider splitting to two sentences.

• Hypotheses are not clearly defined in the introduction, what are/is your null and alternative hypotheses?

Methods:

• “A periodontitis that was classified as moderate was defined as two adjacent alveoli (AL)” alveoli? That is a typo, you mean attachment loss?

• I understand that you mentioned that covariates were picked in accordance of clinical expertise and research, however, I feel a shortcoming is that periodontitis is more complex than that, there are more environmental, habitual and systemic factors that play a role that were not mentioned or included.

Discussion:

• You use of the word “protective” factor against periodontal disease is a little misleading. Periodontal disease is a very complex disease with a very complex pathogenesis, I would not say depression is a protective factor against periodontal disease against obesity.

• You did mention a couple of mechanisms of why the association between periodontitis and depression is negatively correlated in obese patients, however, I think you need to stress more on these mechanisms, because your results are only looking at retrospective data and patient questionnaires. Again, periodontitis is pretty complex and has a complex pathogenesis, this needs to be emphasized in introduction and discussion.

6. PLOS authors have the option to publish the peer review history of their article (what does this mean?). If published, this will include your full peer review and any attached files.

Reviewer #1: **Yes: **Omar Al-karadsheh

Reviewer #2: **Yes: **Hawra AlQallaf, BDM, MSD

---

## [Author Response · Author response to Decision Letter 0]

28 Nov 2024

Response to Isabel Cristina Gonçalves Leite：

1.A rebuttal letter that responds to each point raised by the academic editor and reviewer(s). You should upload this letter as a separate file labeled 'Response to Reviewers'.

This document details our responses to each point raised by the reviewers and the editor. For any suggested revisions, we have outlined our corresponding changes to the manuscript or explanations where necessary.

2.A marked-up copy of your manuscript that highlights changes made to the original version. You should upload this as a separate file labeled 'Revised Manuscript with Track Changes'.

A version of the manuscript containing tracked changes is included to clearly highlight all modifications made in response to the reviewers’ comments.

3.An unmarked version of your revised paper without tracked changes. You should upload this as a separate file labeled 'Manuscript'.

A clean, unmarked version of the revised manuscript is provided for your convenience.

Response to Journal Requirements：

1.Please ensure that your manuscript meets PLOS ONE's style requirements, including those for file naming.

We have carefully reviewed and ensured that the manuscript adheres to PLOS ONE’s formatting requirements, including file naming conventions.

If there are still any formatting issues, please let us know, and we will promptly address them.

2.Please state what role the funders took in the study.

Thank you for pointing out the required amendment to the funding statement. We have updated the statement as follows:

“This study was funded by the Health Commission of Hebei Province (20240102). The funders had no role in study design, data collection and analysis, decision to publish, or preparation of the manuscript.”

This revised funding statement has also been included in our cover letter. Please inform us if further revisions are needed.

3.Please note that your Data Availability Statement is currently missing the repository name.

We noted the absence of the repository name in the Data Availability Statement and have corrected it as follows:

"Publicly available datasets were analyzed in this study, specifically from the National Health and Nutrition Examination Survey (NHANES). These datasets can be accessed at www.cdc.gov/nchs/nhanes/."

4. We note you have included a table to which you do not refer in the text of your manuscript. Please ensure that you refer to Table 1 in your text; if accepted, production will need this reference to link the reader to the Table.

We have reviewed the manuscript and added a reference to Table 1 in the main text to ensure consistency between the table and the manuscript content. This change has been highlighted in red text in the 'Revised Manuscript with Track Changes' document.

Response to “ 5. Review Comments to the Author”：

Reviewer #1: 

1. Explanation of Unexpected Results:

First and foremost, we extend our heartfelt thanks for your meticulous review of our manuscript and the insightful suggestions you have offered. In response to your feedback, we acknowledge the need for a more thorough discussion on how obesity impacts periodontitis through adipokines, particularly in the discussion of adipokines such as leptin and adiponectin. Consequently, we have re-examined a wealth of literature and have added references to adipokines, including leptin and adiponectin, in the fifth paragraph of the discussion section.

2. Consideration of Additional Comparison Groups:

Thank you for your insightful comments on our manuscript titled "Association Study of Depression and Periodontitis in an Obese Population: Analysis Based on NHANES Data from 2009 to 2014." We appreciate your recognition of the focused nature of our study population, which is indeed centered on individuals with obesity.

We acknowledge that our targeted research design may limit the generalizability of our findings. However, our choice to focus on this specific demographic is deliberate, given the established links between obesity and a myriad of health issues, including depression and periodontitis, as well as the growing global prevalence of obesity. Our aim is to provide deeper insights into this high-risk group and to contribute targeted data for future research and clinical practice.

We are grateful for your constructive suggestion and are committed to refining our participant categorization in forthcoming studies to more precisely elucidate the relationship between depression, obesity, and periodontitis. In our next research phase, we will aim to implement more detailed groupings to better understand the complex dynamics at play. Your feedback is invaluable in helping us shape our research approach, and we are dedicated to clarifying the interconnections between these health conditions as per your recommendations.

We appreciate your expertise and look forward to incorporating these insights into our future work.

3. Terminology Consistency:

We appreciate your keen observation regarding the terminology used in our study. We agree that it is essential to maintain consistency in referring to participants as having "depressive symptoms" rather than "depressed" to avoid implying a clinical diagnosis. We have updated our document to reflect this change, replacing all instances of "depressed symptoms" and "depression symptoms" with "depressive symptoms." Thank you for your constructive suggestion.

4. Limitations of the PHQ-9 Questionnaire:

We sincerely appreciate your insightful comments. We have addressed the limitations of PHQ-9 in capturing transient emotional states rather than chronic depressive disorders in the seventh paragraph of the discussion. Your suggestion is highly valuable, and we acknowledge the complexity of diagnosing depression. We will certainly consider incorporating longitudinal measures and clinical diagnostic tools in future studies to more accurately differentiate between temporary and chronic depressive conditions. Your guidance is greatly beneficial for our ongoing research endeavors.

5. Context in Existing Literature:

Thank you for your comments; it appears that you haven't raised any issues in this section.

6. Behavioral Factors and Health-Seeking Assumptions:

Thank you for your insightful comments regarding the assumptions made in our study regarding behavioral factors and health-seeking behaviors. We acknowledge that our current analysis does not directly support the behavioral assumptions due to the lack of relevant data in the NHANES database. We appreciate your guidance on this matter.

In response to your concern, we agree that future studies should delve deeper into this area. We plan to include data collection on participants' health-seeking behaviors, mental health management, and oral hygiene practices to better substantiate our claims. This will allow us to provide a more comprehensive understanding of the relationship between mental health and oral health.

Thank you again for your valuable feedback, which has illuminated a clear direction for our future research endeavors.

7. Unexplored Biological Pathways:

Thank you for your insightful comments. We appreciate the suggestion to explore the metabolic changes in obesity and their impact on inflammatory responses to depression. We will incorporate physiological markers like cortisol, cytokines, and adipokines in our future studies to better understand the interaction between depressive symptoms and periodontal health in the context of obesity.

Reviewer #2: 

Introduction:

1.Thank you for your feedback. The sentence has been revised and split into two: “Obesity is defined as a condition in which the body mass index (BMI) exceeds 30 kg/m². Its prevalence has significantly increased worldwide. In 2008, 1.5 billion adults were classified as overweight, and by 2016, obesity rates had tripled.”

2.Thank you very much for your valuable suggestion! I have already stated the hypotheses (including the null and alternative hypotheses) in the last paragraph of the introduction, but the presentation may not have been sufficiently clear or explicit. I have now further emphasized and refined the statement of the hypotheses in the fourth paragraph of the introduction to ensure that readers can fully understand the research logic. Once again, I sincerely thank you for pointing this out and helping me improve the structure and clarity of the paper.

Methods:

1.Thank you for your careful observation! You are absolutely correct — it was a typo, and I did mean "attachment loss" instead of "alveoli." This error has been corrected in the “Outcome Variable” section of the Materials and Methods. I sincerely appreciate your attention to detail and your valuable feedback. Thank you!

2.Thank you very much for your valuable feedback! As you pointed out, the influencing factors of periodontitis are indeed complex and multifaceted. Due to the limitations of the NHANES database and practical constraints, it was not possible to include all potential confounding factors in this study. We have revised the discussion of the study's limitations accordingly in the seventh paragraph of the "Discussion" section. Thank you again for your insightful suggestions!

Discussion:

1.We sincerely appreciate your detailed feedback regarding the wording in our manuscript. We have thoroughly reviewed the Discussion section and revised the expression of "protective factor" to eliminate any potential misinterpretation. Instead, we now describe the relationship as a "negatively correlated with" in discussion sections. We believe this adjustment more accurately reflects our findings and avoids any misunderstanding. Thank you again for your rigorous review and professional advice, which have been invaluable in helping us improve the manuscript.

2.Based on your suggestions, we have made revisions and additions to the Introduction and Discussion sections of the manuscript. In these revisions, we more thoroughly analyzed the complex pathogenesis of periodontitis and provided a detailed discussion of the potential mechanisms underlying the negative correlation between periodontitis and depression in obese patients. Additionally, we emphasized the retrospective nature of the data and the limitations associated with questionnaire-based research in the Introduction, clearly stating their possible impact on the study results to ensure transparency. 

We hope these revisions meet your expectations and further improve the quality of the manuscript. Thank you again for your thoughtful and constructive comments.

---

## [Editor Report · Decision Letter 1]

1 Dec 2024

Association Study of Depressive Symptoms and Periodontitis in an Obese Population: Analysis Based on NHANES Data from 2009 to 2014

PONE-D-24-44796R1

Dear Dr. shuning Li,

We’re pleased to inform you that your manuscript has been judged scientifically suitable for publication and will be formally accepted for publication once it meets all outstanding technical requirements.

Kind regards,

Isabel Cristina Gonçalves Leite

Academic Editor

PLOS ONE

---

## [Editor Report · Acceptance letter]

8 Dec 2024

PONE-D-24-44796R1 

PLOS ONE

Dear Dr. Li, 

I'm pleased to inform you that your manuscript has been deemed suitable for publication in PLOS ONE. Congratulations! Your manuscript is now being handed over to our production team.

Kind regards, 

on behalf of

Dr. Isabel Cristina Gonçalves Leite 

Academic Editor

PLOS ONE